# Development of an Interprofessional Psychosocial Interventions Framework

**DOI:** 10.3390/ijerph20085495

**Published:** 2023-04-13

**Authors:** Grace Branjerdporn, Kerri Marie Gillespie, Alex Dymond, Neil Josen Delos Reyes, Julia Robertson, Alice Almeida-Crasto, Shailendhra Bethi

**Affiliations:** 1Mental Health and Specialist Services, Gold Coast Hospital and Health Service, Gold Coast 4215, Australia; 2School of Clinical Sciences, Queensland University of Technology, Kelvin Grove 4059, Australia

**Keywords:** mental health, evidence-based practice, capability framework, therapies, psychosocial, training

## Abstract

To meet the increasingly complex needs of mental health consumers, it is essential for multidisciplinary clinicians to have capabilities across a range of psychosocial interventions. Despite this, there is scant evidence investigating the existing levels of knowledge and skills of specialties within multidisciplinary mental health teams. The purpose of this paper was to describe the self-reported capabilities of mental health clinicians, and to provide a rationale for the Psychosocial Interventions Framework Assessment (PIFA), which aims to enhance the access to, and quality of, evidence-informed practice for consumers of mental health services (MHSs) by strengthening workforce capabilities and leadership for psychosocial therapies. Using the Delphi method, the team developed a 75-item survey based on the 10-point Mental Health Recovery Star (MHRS). Participants completed a self-administered survey indicating their perceived capabilities in the PIFA items. The findings revealed lower-than-expected average scores between ‘novice’ and ‘proficient’, highlighting the need for further development of specific training and education modules for individual teams. This is the first framework of its nature to use the Recovery Star^TM^ to determine the psychosocial areas and domains for the assessment of practitioners’ strengths and needs for skill development.

## 1. Introduction

The use of multidisciplinary teams in mental health is the recommended approach for providing comprehensive and optimal care for consumers seeking treatment for mental health issues [1]. The degree and consistency of skills and knowledge within mental health care roles lack research scrutiny. The individual specialties operating within mental health care are subject to varied educational and training requirements, but must perform similar roles and adhere to the same national practice standards [2]. Outside of these standards, there are discipline-specific examples such as ‘Practice Standards for Mental Health Social Workers 2014’ [3], and also clinically specific standards such as the Australian and New Zealand Academy for Eating Disorders clinical practice and training standards [4]. However, it remains that a comprehensive and internationally generalisable template of foundational knowledge for the broader mental health professional base does not exist to provide universal assessment and intervention tools. The development of a framework to identify and measure the core skills required in mental health care would allow for targeted education and training and improved services and outcomes for consumers.

## 2. Background

Mental illnesses, such as depression, anxiety, or schizophrenia, lead to high levels of morbidity and mortality, with significant social and financial costs to society. The 2010 Global Burden of Disease study named mental health and substance use disorders as the leading causes of non-fatal burden of disease, accounting for an estimated 7.4% of total disease burden [5]. In 2010, the direct and indirect costs of mental health disorders were estimated to be greater than AUD $3.6 trillion worldwide, with this figure expected to double by 2030 [6]. Health service utilisation is comparatively high among those suffering mental illness, accounting for 6.4% of all overnight hospitalisations and 3.8% of emergency department (ED) presentations in 2019–2020 [7]. Health policy relating to the treatment of mental health has traditionally been guided by mortality figures, potentially leading to underestimation of, and under-investing in, the overwhelming non-fatal impacts of mental health issues [8].

Biomedical approaches alone are often insufficient to target the diverse range of needs presented by individuals with mental illness [9]. To support these patients, the mental health workforce must be equipped with a broad range of evidence-based medical and psychosocial assessments and interventions. This will ensure the delivery of contemporary, efficient, and recovery-oriented mental health services, as well as optimal outcomes for the consumer and their family [10]. Psychosocial interventions for people with mental illness refer to a broad range of nonpharmacological treatment options, including psychological therapies and social and vocational interventions that aim to improve the quality of life for consumers and their families. Psychosocial interventions have proven effective at improving outcomes for people with depression and schizophrenia, and can lead to a reduction in hospital re-admissions and improved social and vocational functioning [11,12].

Multidisciplinary teams in mental health are standard models of care in Australia, and are recommended as best practice [2]. They generally comprise medical (psychiatrists and psychiatric nurses) and allied health (social workers, psychologists, and occupational therapists) specialties. This combination of expertise is considered optimal to enable the delivery of comprehensive care and improved outcomes for patients with complex needs [1]. Furthermore, having such a team with experience and knowledge across professions (a transdisciplinary approach) allows greater service efficiency, cost effectiveness, less communication confusion for the consumers and families, the development of more holistic and coherent intervention plans and also for the professional development and insight of individual clinicians [13]. Mullen [14] mentioned that this is particularly important in stressful and busy practice environments, such as acute mental health wards where there is a culture of professional separation; in these environments, enhancement of the broader nursing team in a multitude of psycho-social interventions not only leads to improved clinical outcomes, but also to more staff satisfaction and increases in morale. However, the characteristics, roles, competences, and benefits of these collaborative multidisciplinary models of care have received little investigation. The role of ‘mental health clinician’ can be filled by any of a number of healthcare specialties (including psychologists, nurses, dieticians, physiotherapists or occupational therapists). A criticism of this model of care is that each of these specialties have varying levels of training and clinical focus, without a universal framework of required skills, knowledge and clinical standards.

Psychiatrists have the most comprehensive training, i.e., an undergraduate degree or two-year pre-med equivalent, a four-year medical degree, and a minimum one year of general medical training, followed by 60 months of postgraduate training, to become eligible for Fellowship of the Royal Australian and New Zealand College of Psychiatrists [15]. Psychologists and psychiatric nurses must both obtain a Bachelor’s degree. Psycholo-gists must obtain a postgraduate education followed by one or two years of supervised specialist training depending on the length of the post graduate qualification, while nurses may undertake an 8- or 12-subject Graduate Diploma or Master’s in Mental Health Nursing. Other allied health specialties, such as social work and occupational therapy, require only an undergraduate degree with no further training required to work in mental health. The National Practice Standards for the Mental Health Workforce 2013, and discipline-specific practice standards, outline the knowledge and skills required for these mental health professions [2]. These practice standards and other core skills are considered foundational for all specialties implementing patient care in mental health.

This study aimed to identify the pertinent psychosocial interventions, and related skills, that would be considered core knowledge for a mental health clinician. For this purpose, the team members developed the first measure of foundational psychosocial interventions based on the Recovery Star^TM^ [16] to create a framework for education and practice. Mental Health Recovery Star (MHRS) is a multifaceted 10-item outcome measurement and key-working tool that has been widely adopted by service providers in the UK [17]. The Star measures ten psychosocial domains relating to correlates of mental health wellness such as managing mental health, relationship, self-care, addictive behaviour, living skills, social networks, service responsibilities, work, relationships, identity and self-esteem, and trust and hope. The MHRS has high internal consistency and appears to measure an underlying recovery-oriented construct. It has been shown to have good statistically significant item responsiveness, and no obvious item redundancy [17]. The study then aimed to investigate the level of competence and understanding of these core skills in healthcare specialties a public mental health department. The hypothesis of this study was that specialists would report higher levels of ability in skills closely correlated with their educational or employment background, highlighting the benefit of a framework that could be used to identify areas of low competence for targeted education and resource allocation.

One way to measure the efficacy of staff in performing a task is to rate self-perceived competency. While past research has demonstrated contradictory evidence concerning the correlation between self-perceived competence and objectively measured abilities, there is evidence that self-perceived abilities are higher for those that more frequently perform a task [18,19,20,21]. For this reason, it was expected that a self-report measure of core psychosocial interventions would elicit insight into the levels of utilisation and understanding of these therapeutic skills involved in multidisciplinary mental health care. Interventions with low scores would indicate low usage, and/or low confidence, and therefore the need for further training and promotion of specific interventions.

## 3. Materials and Methods

### 3.1. Study Design

The study employed a cross-sectional survey design, collecting data from June 2018 to February 2019.

### 3.2. Participants

Multidisciplinary clinicians (e.g., medical staff, nursing staff, allied health staff, indigenous liaison officers, peer workers, and carer consultants) at a tertiary, public, mental health service in Queensland, Australia were invited to participate. Students undertaking a placement with the mental health service were excluded.

### 3.3. Setting

The mental health service studied is the largest publicly-funded, mental health hospital and health service in the state. The service has community and inpatient mental health teams that are broadly divided into community adult and older persons, community children and youth, inpatient adult and older persons, inpatient children and youth, as well as alcohol and others. Various teams provide a variety of specialised care focused on disorders (e.g., eating disorder program, early psychosis team), age-groups (e.g., adolescent outreach team, perinatal and infant mental health services), and acuity (e.g., continuing care team, extended treatment team).

### 3.4. Ethical Considerations

This study was exempt from ethical review by the Gold Coast Hospital and Health Service Ethics Committee: LNR/2018/QGC/47513.

### 3.5. Tool Development

To develop the Psychosocial Interventions Framework Assessment (PIFA), and reach a consensus on the psychosocial interventions included in the questionnaire, the Delphi method [22] was used. The Delphi method is a process of identification; panel selection; and iterative communication to reach a consensus where knowledge in uncertain. Focus groups were employed because this method enables interactions, generates discussion about a range of topics, and encourages participants to make explicit their clinical reasoning. To first identify all potential interventions and frameworks, a comprehensive literature review was conducted, along with benchmarking of other health services concerning similar frameworks. Rigorous consultation was completed through focus groups with mental health experts on the knowledge, skills, and attitudes required of a mental health workforce. Focus groups were facilitated by two clinicians and the participants were asked about the psychosocial interventions that they considered as core for mental health professionals across the service. A board and representative group of mental health experts were purposively selected from the Gold Coast Hospital Mental Health and Specialist Services (MHSS). The final sample included 40 participants such as consultant psychiatrists and senior clinicians, psychologists, occupational therapists, social workers, peer workers, team leaders, and service directors. The draft survey was reviewed across 12 h long meetings in a six-month iterative process to ensure consultation from key stakeholders.

### 3.6. Measures

Psychosocial Interventions Framework Assessment (PIFA): The PIFA is a self-report measure that has 75 items reflecting psychosocial interventions considered core to mental health staff, developed around the ten domains of the MHRS: managing mental health; physical health and self-care; living skills; social networks; education and work; relationships; addictive behaviours; responsibilities; spirituality, identity and self-esteem; and trust and hope [16]. The MHRS is a ten-point model that was developed by Triangle Consulting and the Mental Health Providers Forum [16,23]. This model was designed to be given to mental health service users, and measures recovery and change over these ten domains. Each of these domains contains between 3 and 17 related psychosocial interventions (see Table 1 for details). Based on the level of capability, items are scored on a four-point scale from 1 = “minimal capability”; 2 = “novice capability”; 3 = “proficient capability”; to 4 = “expert capability”.

### 3.7. Statistical Analysis

Data were analysed using IBM SPSS, version 27.0 [24]. Descriptive statistics (e.g., average, standard deviation, and frequency) were computed for demographic variables, and PIFA items as a whole sample and for each discipline.

## 4. Results

Of the 98 clinicians who completed the questionnaire, 22 (22.4%) were nursing staff, 12 (12.2%) were medical staff, 12 (12.2%) were occupational therapists, 13 (13.3%) were psychologists, 17 (17.3%) were social workers, and 11 (11.2%) were unknown. The remaining staff (*n* = 11, 11.2%) came from a range of backgrounds such as peer support workers, administration officers, pharmacists, Indigenous officers, speech pathologists, and recreation officers. The majority of respondents (*n* = 61, 62.2%) reported working full-time employment (FTE), while 20 (20.4%) reported working 0.5–0.8 FTE, and 4 (4.1%) reported 0.2–0.4 FTE. Clinicians had worked in mental health for less than 2 years (*n* = 16, 16.3%); 3–4 years (*n* = 8, 8.2%); 5–10 years (*n* = 22, 22.4%); 11–20 years (*n* = 29, 29.6%); or over 20 years (*n* = 12, 12.2%), with 11 (11.2%) not answering this question. Results for all domain scores are described in Table 1.

The average scores for all professions in each domain were between ‘novice’ (2) and ‘proficient’ (3). Detailed scores for each domain and intervention, for all respondents and by individual specialty, can be viewed in Table 1. Managing mental health comprised items related to psychoeducation and psychopathology assessment and treatment. Psychology scored the highest on this domain overall, followed by medical staff, with occupational therapy scoring themselves the lowest. All staff members scored themselves closer to ‘minimal capability’ (1) or ‘novice capability’ (2) for seclusion and Restraint, Electroconvulsive Therapy (ECT) and Transcranial Magnetic Stimulation (TMS), and Eating Disorder Psychoeducation.

Physical Health and Self-Care was the lowest-rated domain, with psychology and social workers considering themselves the least capable in this area. Nurses had the highest capability score; midway between ‘novice capability’ (2) and ‘proficient capability’ (3). All staff scored their lowest scores for Bowel and Bladder Management and Sexual Health. Occupational therapy scored highest overall for the Living Skills domain, which focuses on daily function, accommodation, community engagement, and daily tasks. Every specialty scored Music Therapy as the intervention where they felt least capable.

Social Networks was the highest-rated domain for all respondents, although only psychology scored an average domain score above ‘proficient capability’ (3). The Work and Education domain was the lowest rated by medical staff. Psychology scored highest on the domain of Relationships, and on each individual intervention it contained. Occupational therapy rated themselves the least capable in this particular domain. For the domain of Addictive Behaviours, all specialties rated themselves an average score of below ‘proficient capability’ (3) but greater than ‘novice capability’ (2).

Responsibilities included an understanding of relevant legal requirements and duties, ethical responsibilities, social services, and evaluating capacity, well-being, and quality of life. This was the second lowest rated of all domains. The highest-rated intervention here was Clinical Outcomes Evaluation, with the lowest being National Disability Insurance Scheme (NDIS) Psychoeducation. Nurses and medical staff felt the most capable regarding the Mental Health Act 2016 [25], while psychology and social workers rated themselves more capable regarding the Child Safety and Protection Act 1999, and Ensuring Human Rights.

Spirituality, Identity, and Self-Esteem included tailored care to minority and marginalised groups, and building resilience. This was a comparably low-scoring domain, with only psychology rating themselves over ‘proficient capability’ (3) for any intervention. The final domain was Trust and Hope, which comprised trauma-informed care, family centred-care, motivational interviewing, and therapeutic alliance. Psychology workers rated themselves more highly on this domain, and on each intervention. All specialties rated themselves at least ‘proficient’ on average (over 3) for Therapeutic Alliance.

## 5. Discussion

This is the first paper of its kind to outline the psychosocial areas that pertain to each domain in the Recovery Star^TM^ [16], a comprehensive framework that has been developed for the assessment of capability as well as direction for further training. Using this framework, the paper investigated the self-reported capabilities of mental health clinicians regarding core psychosocial interventions outlined in the National Practice Standards for the Mental Health Workforce 2013, and identified by a mental health clinician peer group [2]. The results saw similarly low–average scores across disciplines, but higher scores were more indicative of respondents’ specialisation-specific skills and roles.

The Managing Mental Health domain was the largest domain, consisting of interventions related predominantly to the direct and immediate impacts and requirements of mental health pathologies. The workforce rated themselves as sitting in between ‘novice’ and ‘proficient’ capability for this domain. Medical staff and psychologists scored highest on this domain, with occupational therapy scoring the lowest on almost all interventions. This is an unsurprising finding given the extent of psychology and psychiatry education devoted to psychopathology diagnosis and treatment. However, there is an inherent need for optimum psychoeducation for both the mental health workforce and clientele, especially on common mental health presentations, such as anxiety and depression, for more appropriate diagnosis, prognosis, and treatment [26]. An understanding of psychopathology must also be paired with the education on dual-diagnosis and co-morbidities, as those suffering from such conditions may require an inter-disciplinary approach to treatment [27]. Self-rated capacity around eating disorders and ECT/TMS was low among all specialty groups; however, these are specialist areas with specified staff dedicated to treatment and care.

Medical staff, followed by nursing, scored the highest for Medication Psychoeducation. This is not an unusual finding given doctors’ sole prescribing authority, and the role of nurses in medication distribution. The higher score from nursing for Risk Assessment may reflect the reliance on nursing staff to conduct mental health triage, or triage in general as part of the nursing role [28]. The assessment of risk of a patient to cause harm to themselves or others is essential to the provision of care in mental health [29]. While participants scored comparatively higher for this intervention, scores below at least ‘proficient’ (3) imply a lack of clinician knowledge, practice, or confidence, thereby posing a potential risk to consumers, staff, and the public.

Seclusion and Restraint are regulated under Queensland’s Mental Health Act 2016 [25], which states that these interventions must be approved by the Chief Psychiatrist under strict criteria. Medical staff, therefore, should have an in-depth understanding of this form of treatment given their legal obligations. All allied health professions, particularly that of occupational therapy, rated themselves comparatively low (closer to ‘novice’). This likely indicates an intervention that is not commonly used, as it is used by frontline workers as a last resort in de-escalation of high-risk contexts which correlates to the patient and staff trauma. However, given the potential for trauma and adverse patient outcomes, this knowledge gap is of particular concern amongst the allied health representation.

The Physical Health and Self-Care domain considers lifestyle management, including some suggestions on healthy habits which may increase overall health and well-being. Maintaining a good diet, sleep, and exercise regimen have been shown to improve biological and cognitive health and avoid some mental health presentations, such as depression [30]. This domain was rated the poorest across the workforce, scoring on the lower end between novice and proficient. These results are concerning given the generalist health background for nursing and medical professions, and the significant role of this domain in the long-term care of chronically unwell psychiatric patients [31]. People with sexual, bladder, and bowel issues have multiple quality of life and mental health struggles [32,33,34]. Surprisingly, medical staff only rated themselves between novice and proficient for cognitive function, which was lower than psychology and equal to occupational therapy. This may indicate that psychologists perform more of the cognitive testing than psychiatry registrars, or that medical staff lack confidence in these areas and require refresher training. Without concurrent objective skills and measurements, this is difficult to determine.

The Living Skills domain provides opportunities for finding creative, personal, interpersonal, or even just divertive outlets to potentially reduce stress, as well as gaining an understanding of life-affecting developmental milestones and social issues. Apart from stress relief strategies, this domain also touches on understanding one’s developmental milestones which, when understood well, may help avoid the potential progression of some psychopathologies which can occur later in life [35]. This domain also addresses homelessness, which is associated with significant mental health presentations following its occurrence [36]. Occupational therapy rated the highest in this domain; however, the workforce overall rated themselves closer to novice. Occupational therapists self-rated between proficient and expert on a number of interventions in this area, including domestic tasks, community engagement, coping skills, sensory modulation, and activity scheduling. This is congruent with these domains relating to the Functional Assessment role of occupational therapy in psychiatric rehabilitation [37]. Accommodation and Homelessness were scored between novice and proficient across all disciplines, except medical, who ranked themselves below ‘novice capability’ (2). Given the lack of homelessness services on the Gold Coast, and the dominant role this has for inpatient social workers [38], it would be beneficial for other professions to assist in this area, particularly psychology workers, who rated themselves remarkably close in capacity to social work.

The Social Networks domain offers support for numerous social life aspects, including improving social skills, informal networking and service engagement, and considering a systemic approach for acknowledging all stakeholders. Good social skills can result in the formation of robust social supports, which reportedly prevents the mental health effects of not having any to begin with [39]. Establishing an informal network of support may also promote health-favouring behaviors [40]. Strong service engagement reportedly allows for optimised resolution of some mental health presentations, such as depression [41]. This domain had the highest-rated self-competence across the workforce. All professions rated themselves closer to proficient than novice in this area, except psychology workers, who rated themselves between proficient and expert, including the areas of service coordination and family engagement. These roles in the active care coordination are traditionally placed on other professions, such as social workers [42]. Given that multidisciplinary teams can often be characterised by an uneven distribution of workloads, causing significant dynamic and functional issues [43], this is perhaps an area psychology workers can be more involved in.

The Work and Education domain comprises three interventions aimed at supporting and educating consumers in these areas. Obtaining support for vocational up-skilling may prevent minor depression presentations, especially in older, long-unemployed individuals [44]. Beyond supporting those undertaking potentially stressful pursuits, assisting these individuals in finding enjoyable leisure activities could also improve their wellbeing [45]. This domain scored in-between ‘novice capability’ (2) and ‘proficient capability’ (3) across the workforce. Medical staff scored lowest in this area, with occupational therapy and psychology scoring highest. Social work and occupational therapy have traditionally been professions that have had the most research and investment in employment and employment programs [46,47,48]. It is vital for both the fields of psychology and psychiatry to be involved in this area of psychosocial intervention, as evidence for impairment to work and allotment of a disability support pension requires their input [49].

The Relationships domain combines psychoeducation, management, and support paradigms on the different relationships an individual may engage in. Domestic and Family Violence (DFV) Support was one intervention that saw relatively low scores for all specialties, particularly for nursing, medical and occupational therapy, who all rated themselves closer to ‘novice capability’ (2). Whilst psychology has a core interest in relationships, it is still interesting that psychology workers rated themselves higher in areas such as DFV, which is traditionally the realm of social work [50]. Low scores regarding DFV may indicate they do not feel supported or adequately trained to effectively identify or treat these patients in their current role, or that this form of care is predominantly conducted by a staff specialist. Women who experience DFV often do not seek any form of help, but have an increased risk of suffering mental health issues and higher rates of healthcare presentations [51]. This makes the development of skills for identifying and managing DFV integral to mental health care [52].

The Addictive Behaviours domain comprises the management of physically and psychologically debilitating activities. Addictions may become resolvable using diversified management strategies, such as those used by substance use disorder treatment organisations [53]. Psychology ranked itself higher in this area, possibly reflecting the role of motivational interviewing in addiction. It is important to note that substance use in mental health is linked to increased patient disorderly, and potentially violent, behaviour requiring de-escalation [54]. Addictive behaviours also involve high levels of care and numerous physical comorbidities, all requiring substantial resources and expertise [55].

The study found low average ‘novice capability’ (2) scores given to the legal and ethical aspects of care reflected in the domains of Responsibility and Spirituality, Identity, and Self-Esteem. The Responsibilities domain described the legal and ethical aspects of providing mental health, legal protection, and support to vulnerable populations, including an understanding of child protection laws, and the Mental Health Act 2016. Understanding the terms included in these Acts and guiding principles would be essential for those wanting to provide optimal care for vulnerable consumers while ensuring the protection of their human rights. Social welfare support, such as that provided by Centrelink in Australia, is easily accessible, but may require some assistance for those unable to request these services. Psychology and social workers rated themselves the highest in this domain; however, no specialty rated themselves above ‘proficient’ (3) in any category, with the average score leaning closer to ‘novice’ (2). Psychiatry workers ranked themselves higher in knowledge of the Mental Health Act 2016 and, indeed, they are the cornerstone of the lawful implementation of many of the restrictions and involuntary treatments the Act can provide. Social workers scored higher in knowledge of welfare provision (Centrelink and NDIS), again reflecting their primary role in this area.

The Spirituality, Identity, and Self-Esteem domain acknowledges the diversity in beliefs and ideologies that every individual may have, and then tailors care towards these. Because of the strong links between cultural ethnicity and certain religions, engaging these populations in their respective belief systems may be beneficial for an individual [56]. Care can be tailored towards addressing the cultural and linguistic diversity of different populations, which ensures better efficacy for various mental health treatment options [57]. Addressing the needs of the LGBTQIA+ populations may also improve the overall response to treatment of those who identify in the community and who suffer from mental health presentations [58]. Social workers ranked themselves higher than others in this area, but given the individualised nature of identity, self, and spirituality, it would be difficult to quantify this knowledge. Perhaps this belief stems from social work’s core values of empowerment, self-determination, and person-centred approaches [59].

The last domain of Trust and Hope features numerous interventions which focus on various aspects of upholding mental health. It was ranked highest by psychology workers, with most categories ranked between proficient and expert. It should be noted that all professions ranked themselves highly in this area, particularly in Therapeutic Alliance, which is the cornerstone of professional mental health recovery outcomes [60]. Therapeutic alliance strongly aligns with person-centered care planning, which may optimise mental health care provision for consumers [61].

In the study setting, most patients are under the Mental Health Act 2016 [25], where administrative responsibility and liability falls to psychiatry workers. It is perhaps this statutory responsibility and liability that makes psychiatry workers hesitant to score themselves highly outside their traditional domains, congruent with modern defensive medicine [62], and may also come from the medical principle of shared decision-making with fields of non-expertise [63]. Occupational therapy in a multi-disciplinary team is defined by its strength of identity and knowledge of its professional boundaries and capabilities [64]. Indeed, this can be seen in the current findings of occupational therapists consistently ranking themselves lower compared with others in non-traditional occupational therapy domains, whilst viewing themselves favourably in those traditionally correlated. Social workers, on the other hand, can often view themselves as holistic practitioners, leaving themselves open to the blurring of professional roles correlating with the allocation of non-social-work-related tasks. This can result in increased workloads further diminishing the opportunity to set their professional identity [43,64].

## 6. Limitations

The study was conducted at one site only, in a nursing-dominant cohort. This may limit the generalisability of the findings. It is recommended that this framework be utilised in a wider network in the future to identify fundamental skills, areas, or clinician groups that would benefit from strategies such as increased training, in-service education, or online educational modules and resources. There is also some evidence that healthcare professionals may over- or under-estimate their abilities and competence levels, and may feel more competent in activities they perform more frequently, independent of skill [21]. For this reason, it would be optimal to capture objective, measurable data on abilities, as well as determine the frequency of intervention utilisation to compare with the results of the PIFA for a more comprehensive understanding of multidisciplinary competence in foundational interventions and skill types.

## 7. Conclusions

In general, specialists rated themselves more capable in areas closely related to their field of expertise. This supports the findings of previous self-report studies; participants rate themselves as more capable of activities they commonly conduct than those they rarely practice [19,21]. Any of these specialists may be employed in the role of ‘mental health clinician’, which requires a sound grasp of all foundational skills and interventions listed in the PIFA. Therefore, these gaps in knowledge pose a risk for inconsistent treatment dependent on clinician specialisation. Higher PIFA ratings do not necessarily indicate a higher level of knowledge or skill [20]; however, it is a good indicator of areas requiring further education, training, or practice. All staff rated themselves far below expert on almost every intervention and domain, including legal acts and requirements that are fundamental to their employment. These findings demonstrate the need for continued research, education, and training in core skills for mental health care, and highlight the benefits of an Interprofessional Psychosocial Interventions Framework to highlight areas of deficiency to improve multidisciplinary care.

## Figures and Tables

**Table 1 ijerph-20-05495-t001:** Self-rated capability for Psychosocial Interventions Framework Assessment (PIFA) domains and interventions.

		Nursing	Medical	Occupational Therapy	Psychology	Social Work	All
Domain							
Managing Mental Health		Mean (SD)	Mean (SD)	Mean (SD)	Mean (SD)	Mean (SD)	Mean (SD)
	Risk Assessment	3.09 (0.75)	2.92 (0.67)	2.67 (0.98)	3.38 (0.51)	2.94 (0.75)	2.96 (0.85)
	Recovery Planning—purpose, care planning	2.86 (0.83)	2.75 (0.87)	3.33 (0.98)	3.23 (0.60)	2.82 (0.95)	2.96 (0.86)
	Depressive Disorder Psychoeducation	2.95 (0.79)	3.17 (0.83)	2.75 (0.75)	3.62 (0.51)	2.76 (0.90)	2.95 (0.85)
	Anxiety Disorder Psychoeducation	2.86 (0.71)	3.17 (0.94)	2.83 (0.58)	3.54 (0.52)	2.65 (0.79)	2.93 (0.80)
	Psychotic Disorder Psychoeducation	2.95 (0.72)	3.25 (0.87)	2.92 (0.90)	3.08 (0.64)	2.71 (0.99)	2.87 (0.89)
	De-escalation techniques	2.91 (0.81)	2.67 (0.78)	2.50 (0.80)	3.42 (0.51)	2.76 (0.83)	2.84 (0.83)
	Undertaking Suicide Prevention Pathway	2.77 (0.87)	2.83 (0.83)	2.50 (0.90)	3.23 (0.60)	2.56 (0.96)	2.72 (0.89)
	Bipolar Disorder Psychoeducation	2.77 (0.81)	3.17 (0.83)	2.75 (0.87)	3.15 (0.55)	2.53 (0.94)	2.72 (0.91)
	Understanding Dual-Diagnosis and Co-morbidities	2.77 (0.81)	2.75 (1.06)	2.25 (0.87)	3.00 (0.82)	2.82 (0.81)	2.64 (0.90)
	Trauma Psychoeducation	2.45 (0.74)	2.58 (1.00)	2.25 (0.75)	3.38 (0.51)	2.75 (0.77)	2.62 (0.84)
	Personality Disorder Psychoeducation	2.73 (0.77)	2.50 (0.90)	2.67 (0.49)	3.15 (0.55)	2.24 (0.75)	2.58 (0.84)
	Medication Psychoeducation	2.82 (0.59)	3.33 (0.65)	1.92 (1.00)	2.69 (0.63)	2.25 (0.86)	2.49 (0.94)
	Attachment-Related Disorder Psychoeducation	2.45 (0.74)	2.42 (1.00)	1.67 (0.65)	3.00 (0.82)	2.65 (0.79)	2.46 (0.86)
	Contingency Management	2.64 (0.79)	2.50 (1.09)	2.17 (0.83)	2.62 (0.77)	2.47 (0.80)	2.37 (0.86)
	Eating Disorder Psychoeducation	2.32 (0.84)	2.36 (1.03)	1.67 (0.65)	2.23 (0.93)	2.38 (0.81)	2.20 (0.85)
	Appropriate use of seclusion and restraint (if applicable)	2.50 (0.86)	2.75 (0.87)	1.36 (0.50)	2.25 (1.06)	2.18 (0.81)	2.19 (0.90)
	ECT and TMS Psychoeducation	2.14 (0.77)	2.09 (0.70)	1.33 (0.49)	2.00 (0.82)	1.76 (0.83)	1.79 (0.78)
Physical Health and Self Care		2.52 (0.68)	2.33 (0.73)	2.28 (0.57)	2.41 (0.45)	2.16 (0.74)	2.31 (0.65)
	Sleep Hygiene Strategies	2.91 (0.81)	2.75 (0.97)	2.92 (0.51)	3.31 (0.48)	2.47 (0.94)	2.80 (0.83)
	Cognitive Functioning	2.50 (0.67)	2.58 (1.00)	2.58 (0.67)	3.15 (0.55)	2.35 (1.00)	2.53 (0.86)
	Management of Co-morbid Health Conditions	2.50 (0.86)	2.92 (1.00)	2.42 (0.79)	2.85 (0.99)	2.29 (0.92)	2.51 (0.93)
	Physical Activity Management	2.59 (0.91)	2.58 (0.79)	2.50 (0.80)	2.54 (0.52)	2.35 (1.00)	2.49 (0.82)
	Nutrition and Diet Management	2.55 (0.86)	2.33 (0.89)	2.50 (0.80)	2.31 (0.75)	2.24 (0.97)	2.40 (0.86)
	Sexual Health Management	2.27 (0.77)	2.18 (0.75)	1.67 (0.65)	2.00 (0.71)	2.06 (0.90)	2.02 (0.79)
	Mobility Management	2.48 (1.03)	1.58 (0.79)	2.17 (1.03)	1.69 (0.85)	1.88 (0.93)	1.93 (0.94)
	Bowel and Bladder Management	2.36 (0.90)	1.67 (0.78)	1.50 (0.80)	1.46 (0.66)	1.65 (0.86)	1.77 (0.90)
Living Skills		2.56 (0.72)	2.20 (0.64)	2.74 (0.49)	2.64 (0.43)	2.47 (0.64)	2.47 (0.61)
	General Coping Strategies	2.95 (0.84)	2.67 (0.78)	3.17 (0.58)	3.38 (0.65)	2.88 (0.86)	2.97 (0.77)
	Emotional Regulation Support (including anger and distress)	2.82 (0.80)	2.58 (0.79)	2.75 (0.62)	3.46 (0.52)	2.76 (0.83)	2.81 (0.78)
	Activity Scheduling	2.77 (0.92)	2.42 (0.79)	3.17 (0.58)	3.23 (0.60)	2.59 (0.87)	2.76 (0.81)
	Community Engagement	2.77 (0.87)	2.17 (0.83)	3.18 (0.60)	2.62 (0.51)	2.76 (0.97)	2.70 (0.80)
	Understanding Life Stage and Developmental Functioning	2.64 (0.85)	2.50 (1.09)	2.75 (0.87)	3.25 (0.75)	2.71 (0.77)	2.70 (0.87)
	Mindfulness/Relaxation Strategies	2.73 (0.98)	2.58 (0.51)	2.75 (0.45)	3.15 (0.69)	2.59 (0.87)	2.68 (0.80)
	Interpersonal Psychotherapies	2.45 (0.91)	2.58 (1.00)	1.75 (0.75)	3.15 (0.55)	2.53 (0.72)	2.41 (0.88)
	Personal Activities of Daily Living Training (PADLs)	2.71 (1.01)	2.25 (0.97)	3.17 (0.83)	1.92 (0.64)	2.13 (0.96)	2.33 (0.96)
	Accommodation and homelessness	2.27 (0.88)	1.83 (0.72)	2.75 (1.14)	2.23 (0.60)	2.82 (1.01)	2.33 (0.94)
	Sensory Modulation	2.29 (0.85)	1.92 (0.79)	3.25 (0.75)	2.38 (0.65)	2.06 (0.83)	2.32 (0.86)
	Domestic Tasks	2.64 (0.90)	1.83 (0.72)	3.33 (0.65)	1.85 (0.69)	2.29 (0.92)	2.30 (0.93)
	Art/Play Therapy	2.18 (0.80)	1.67 (0.65)	2.08 (0.79)	2.08 (0.95)	2.06 (0.97)	2.01 (0.84)
	Music Therapy Strategies	2.05 (0.72)	1.67 (0.65)	1.58 (0.67)	1.62 (0.65)	1.88 (0.93)	1.74 (0.78)
Social Networks		2.73 (0.85)	2.58 (0.87)	2.69 (0.77)	3.25 (0.43)	2.85 (0.79)	2.78 (0.75)
	Systemic approach	2.77 (1.02)	2.75 (0.97)	2.80 (0.63)	3.38 (0.51)	3.00 (0.87)	2.85 (0.87)
	Social Skills Support	2.68 (0.78)	2.67 (0.89)	2.75 (0.75)	3.23 (0.60)	2.82 (0.95)	2.81 (0.77)
	Support Informal Network Support	2.77 (0.81)	2.67 (0.89)	2.75 (0.87)	3.23 (0.60)	2.71 (0.85)	2.80 (0.77)
	Supporting Service Engagement	2.68 (0.95)	2.25 (0.97)	2.67 (0.98)	3.15 (0.55)	2.88 (0.70)	2.70 (0.86)
Work and Education		2.57 (0.88)	1.92 (0.75)	2.75 (0.71)	2.79 (0.48)	2.57 (0.73)	2.52 (0.75)
	Hobby Psychoeducation	2.67 (0.86)	2.00 (0.85)	2.92 (0.67)	3.00 (0.41)	2.65 (0.79)	2.66 (0.76)
	Educational Support	2.55 (1.00)	1.83 (0.72)	2.67 (0.78)	2.85 (0.55)	2.65 (0.79)	2.49 (0.84)
	Vocational Psychoeducation	2.48 (0.93)	1.92 (0.79)	2.67 (0.89)	2.54 (0.66)	2.41 (0.71)	2.39 (0.80)
Relationships		2.35 (0.70)	2.29 (0.74)	2.08 (0.59)	3.00 (0.59)	2.72 (0.55)	2.44 (0.68)
	Family/Friends/Community/Partner Psychoeducation	2.77 (0.87)	2.75 (0.87)	2.33 (0.78)	3.31 (0.48)	2.94 (0.83)	2.77 (0.80)
	Supporting Engagement with Peer Support Workers	2.55 (0.86)	2.25 (0.87)	2.33 (1.07)	2.92 (0.76)	2.71 (0.77)	2.54 (0.86)
	Grief and Loss Management	2.27 (0.77)	2.42 (0.90)	2.17 (0.58)	3.15 (0.80)	2.59 (0.87)	2.45 (0.83)
	Domestic and Family Violence Support	2.33 (0.73)	2.08 (0.79)	2.08 (0.79)	2.92 (0.64)	2.88 (0.86)	2.42 (0.82)
	Attachment Psychoeducation	2.23 (0.81)	2.42 (0.79)	1.75 (0.62)	3.00 (0.71)	2.71 (0.69)	2.36 (0.82)
Addictive Behaviours		2.69 (0.76)	2.58 (0.89)	2.33 (0.67)	2.92 (0.45)	2.57 (0.79)	2.60 (0.79)
	Harm Minimisation Management	2.71 (0.96)	2.58 (1.00)	2.58 (0.90)	3.15 (0.38)	2.71 (1.05)	2.69 (0.93)
	Smoking Cessation Management	2.81 (0.81)	2.67 (0.89)	2.17 (0.58)	2.85 (0.55)	2.35 (0.93)	2.58 (0.86)
	Substance Use Management	2.57 (0.75)	2.50 (0.90)	2.25 (90.7)	2.77 (0.60)	2.65 (1.00)	2.52 (0.84)
	Other Addictive Behaviour Management	2.33 (0.80)	2.42 (0.90)	1.92 (0.67)	2.83 (0.72)	2.41 (0.94)	2.33 (0.85)
Responsibilities		2.38 (0.69)	2.22 (0.74)	2.32 (0.65)	2.52 (0.41)	2.51 (0.67)	2.32 (0.66)
	Clinical Outcomes Evaluation	2.62 (90.7)	2.25 (0.87)	3.00 (1.04)	3.00 (0.71)	2.53 (0.87)	2.60 (0.95)
	Child Safety/Protection Act 1999	2.62 (0.92)	2.42 (1.00)	2.17 (0.83)	2.92 (0.64)	2.94 (0.75)	2.55 (0.87)
	Mental Health Act 2016	2.90 (0.54)	2.75 (0.75)	2.50 (0.80)	2.62 (0.77)	2.29 (0.85)	2.51 (0.80)
	Ensuring Human Rights	2.57 (0.93)	2.25 (0.87)	2.36 (0.081)	2.62 (0.65)	2.94 (0.97)	2.50 (0.91)
	Capacity	2.48 (0.81)	2.58 (1.00)	2.58 (0.79)	2.62 (0.77)	2.36 (0.84)	2.41 (0.86)
	Safeguarding vulnerable populations	2.43 (0.93)	2.17 (0.94)	2.17 (0.83)	2.85 (0.38)	2.59 (0.80)	2.36 (0.83)
	Independent Patient Rights Advisor Support	2.33 (0.86)	2.00 (0.74)	2.33 (0.78)	2.17 (0.58)	2.18 (0.81)	2.17 (0.78)
	Social Welfare (Centrelink) Support	2.00 (0.71)	1.75 (0.62)	2.00 (0.77)	2.15 (0.55)	2.76 (1.03)	2.10 (0.82)
	Forensic Mental Health	2.10 (0.83)	2.25 (1.06)	1.92 (1.00)	2.31 (0.63)	2.24 (0.83)	2.07 (0.85)
	NDIS ^1^ Psychoeducation	1.76 (0.70)	1.75 (0.75)	2.25 (0.87)	2.00 (0.82)	2.18 (0.88)	1.93 (0.80)
Spirituality, Identity, and Self-Esteem		2.26 (0.82)	2.17 (0.78)	2.30 (0.36)	2.57 (0.57)	2.65 (0.70)	2.36 (0.71)
	Building resilience and self-efficacy	2.57 (0.93)	2.50 (1.00)	2.42 (0.67)	3.15 (0.69)	2.88 (0.86)	2.67 (0.85)
	Support with Engagement with Religion	2.48 (0.98)	2.25 (0.87)	2.33 (0.65)	2.54 (0.88)	2.71 (0.85)	2.43 (0.88)
	Care Tailored to Culturally and Linguistically Diverse Populations	2.19 (0.75)	2.00 (0.74)	2.42 (0.79)	2.31 (0.75)	2.65 (0.79)	2.28 (0.78)
	Care Tailored to Aboriginal and Torres Strait Islander Populations	2.14 (0.91)	2.08 (0.79)	2.25 (0.45)	2.46 (0.66)	2.59 (0.80)	2.28 (0.80)
	Care Tailored to LGBTIQ+ Populations	2.18 (0.91)	2.00 (0.74)	2.08 (0.51)	2.38 (0.65)	2.38 (0.72)	2.17 (0.76)
Trust and Hope		2.52 (0.79)	2.58 (0.76)	2.60 (0.50)	3.29 (0.51)	2.82 (0.66)	2.71 (0.71)
	Therapeutic Alliance	3.00 (0.95)	3.17 (0.58)	3.17 (0.72)	3.62 (0.51)	3.18 (0.64)	3.17 (0.76)
	Motivational Interviewing	2.52 (0.93)	2.92 (0.90)	2.75 (0.87)	3.31 (0.63)	2.47 (0.62)	2.70 (0.88)
	Managing group dynamics	2.25 (0.91)	2.25 (0.87)	2.50 (1.09)	3.15 (0.80)	2.88 (0.86)	2.59 (0.93)
	Trauma-informed Care and Practice	2.43 (0.98)	2.33 (0.89)	2.58 (0.67)	3.38 (0.65)	2.65 (0.79)	2.58 (0.86)
	Family-Centred Care	2.43 (0.87)	2.25 (0.87)	2.00 (0.60)	3.00 (0.71)	2.94 (0.97)	2.49 (0.88)

^1^ National Disability Insurance Scheme.

## Data Availability

The Psychosocial Interventions Framework Assessment, or the data that support the findings of this study are available on request from the corresponding author, G.B. The data are not publicly available due to concerns that information could breach the privacy of the participants.

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
