# Peer review of "Development of an Interprofessional Psychosocial Interventions Framework"

_ijerph, 2023, doi:10.3390/ijerph20085495_

Round 1

Reviewer 1 Report

Thank you for giving me the opportunity to review the manuscript "Development of an interprofessional Psychosocial Interventions Framework" . So the manuscript is appropriate for International Journal of Environmental Research and Public Health.

 It would be interesting to see if there is an effect of Clinicians's tenure

I think the paper is ready for publication

Reviewer 2 Report

The authors have designed and implemented a considered and in-depth research project to examine the education and training levels of mental health teams within a large Queensland hospital.  The literature search and presentation include a number of individual occupations that are included within mental health teams and facilities.  They have questioned whether the varying levels of learning are sufficient to interact as a cohesive treatment team covering all aspects of mental health.  This poses the question of the suitability of individual mental health professional or the need for fewer but broader qualifications and training, that is, less specialisation in each occupational role as is currently the position.

The research method of self-reported surveys of 98 clinicians produces a reliable set of frequencies for descriptive statistics while the development of the survey using the Delphi Technique indicates a valid survey instrument.  The descriptive statistics of mean and standard deviation provide for a good interpretation of the data and interpretation of the data.  The discussion provides an interpretation of these statistics with a comparison of the various factors within each category.  The authors acknowledge the limitations of their research and suggest further research to extend their work.

Overall, this a well designed and implemented research project that will pre-empt further discussion among mental health practitioners with the aim of developing a cohesive multidisciplinary mental health team which should ensure an improved standard of care for mental disorders.

Attached is a copy of the review paper showing suggested alterations for reading clarity and presentation.

Reviewer 3 Report

Thank you for giving me an opportunity to review such a presious paper. I believe that it deals with one of the most important topics in community care. However, there are critical study limitations.

1. The authors focused on "development" of an interprofessional psychosocial interventions framework. They tried to identify the components of knowlege and skills of specialities within multidicsciplinary mental health teams. The methodology is not clear such as: what was interview guide like?, How were the participants' professional experiences?, Where did the participants come from?, How long did the discussion last?, How did they identify the components?, and so on.

2. As the authors described, the study setting is limited to a tertiary, public, ental health service. Is it representative facility? 

3. There is no statistical analysis. It seems not to be scientific. 

4. Is the paper aimed to confirm the differences among health care professionals in capabilities? If so, the authors should reconsider study protocol.

Reviewer 4 Report

Thank you for the opportunity to reviw this article. The work is interesting and also the results reported. However, I think the authors should revise the paper with respect to some points. 

The introduction is unclear and not very straightforward. In some places it seemed a bit confusing to me. I suggest revising it so as to include relevant content with the objective of the paper.

The authors used a Delphi procedure. However they say little with respect to this procedure. I suggest better specifying especially the methodology of using this procedure 

The results seem to have been reported in the form of means and standard deviations. However, I suggest that they also include corsstabs by crossing some variables and report the response rate of the items.

Round 2

Reviewer 3 Report

The paper has been improved enough to be accepted. Especiallly, the methodology section becames clear.